# PD-L1 is expressed on human activated naive effector CD4+ T cells. Regulation by dendritic cells and regulatory CD4+ T cells

**Fabienne Mazerolles**[1,2¤]*, **Frédéric Rieux-Laucat**[1,2¤]

**1** Laboratory of Immunogenetics of Paediatric Autoimmunity, Mixed Research Unit 1163, Institut National de la Santé et de la Recherche Médicale, Paris, France, **2** Imagine Institute Paris, Paris Descartes – Sorbonne Paris Cité University, Paris, France

¤ Current address: Institut National de la Santé et de la Recherche Médicale, Mixed Research Unit 1163, Laboratory of Immunogenetics of Pediatric Autoimmunity - Necker Enfants Malades Hospital, Paris, France
* fabienne.mazerolles@inserm.fr

**Data Availability Statement:** All relevant data are within the manuscript and its Supporting information files.

## Abstract

The T cell expression of various co-signalling receptors from the CD28 immunoglobulin superfamily (Inducible T cell co-stimulator (ICOS), Programmed cell death 1(PD-1), cytotoxic T lymphocyte associated protein 4 (CTLA-4), B and T lymphocyte attenuator (BTLA) or from the tumour necrosis factor receptor superfamily (glucocorticoid-induced TNFR family related (GITR), 4-1BB, and CD27), is essential for T cell responses regulation. Other receptors (such as T cell immunoglobulin and mucin domain-containing protein 3, T cell immunoglobulin and T cell immunoglobulin and ITIM domain (TIGIT), and lymphocyte activation gene 3) are also involved in this regulation. Disturbance of the balance between activating and inhibitory signals can induce autoimmunity. We have developed an *in vitro* assay to simultaneously assess the function of naive CD4+ effector T cells (TEFFs), dendritic cells (DCs) and regulatory T cells (TREGs) and the expression of co-signalling receptors. By running the assay on cells from healthy adult, we investigated the regulation of activated T cell proliferation and phenotypes. We observed that TEFFs activated by DCs mainly expressed BTLA, ICOS and PD-1, whereas activated TREGs mainly expressed TIGIT, ICOS, and CD27. Strikingly, we observed that programmed death-ligand 1 (PD-L1) was significantly expressed on both activated TEFFs and TREGs. Moreover, high PD-L1 expression on activated TEFFs was correlated with a higher index of proliferation. Lastly, and in parallel to the TREG-mediated suppression of TEFF proliferation, we observed the specific modulation of the surface expression of PD-L1 (but not other markers) on activated TEFFs. Our results suggest that the regulation of T cell proliferation is correlated with the specific expression of PD-L1 on activated TEFFs.

## Introduction

A large number of co-signalling molecules are involved in the production of co-stimulatory or co-inhibitory signals in the regulation of T cell activation. The expression of co-signalling molecules on effector T cells (TEFFs) initiates T cell responses, and the expression on regulatory T

**Funding:** The study was funded by the Institut National de la Santé et de la Recherche Médicale (INSERM), a grant from the French government (managed by French National Research Agency (Agence National de la Recherche) as part of the "Investment for the Future" program (Institut Hospitalo-Universitaire Imagine, grant ANR-10-IAHU-01, Recherche Hospitalo-Universitaire, grant ANR-18-RHUS-0010)), and other grants from the Agence National de la Recherche (ANR-14-CE14-0026-01 "Lumugene"; ANR-18-CE17-0001 "Action"), the Fondation pour la Recherche Médicale (Equipe FRM EQU202103012670), the Ligue Contre le Cancer – Comité de Paris, Fondation ARC pour la recherche sur le CANCER, and the Centre de Référence Déficits Immunitaires Héréditaires (CEREDIH).

**Competing interests:** The authors have declared that no competing interests exist.

**Abbreviations:** B7-H1, B7 homolog 1; BTLA, B and T lymphocyte attenuator; CFSE, carboxyfluorescein succinimidyl ester; CTLA-4, cytotoxic T-lymphocyte associated protein 4; CTV, CellTrace™ Violet; DC, dendritic; GITR, glucocorticoid-induced tumour necrosis factor receptor family related; ICOS, inducible T cell co-stimulator; LAG3, lymphocyte activation gene 3; PBMC, peripheral blood mononuclear cell; PD-1, programmed cell death 1; PD-L1/2, programmed death-ligand 1/2; SEE, staphylococcal enterotoxin E; TCR, T cell receptor; TEFFs, effector CD4+ T cells; TIGIT, T cell immunoglobulin and ITIM domain; TIM3, T cell immunoglobulin and mucin domain-containing protein 3; TREGs, regulatory T cells.

cells (TREGs) enables the latter to control TEFF activation [1]. The balance between co-inhibitory and co-activator receptors regulates autoimmune diseases, many of which have been linked to genetic variations in co-inhibitory receptors [2]. The majority of these co-signalling molecules can be classified into two families on the basis of their structure. Firstly, the CD28 immunoglobulin superfamily [3] includes inducible co-stimulatory molecule (ICOS), cytotoxic T-lymphocyte antigen-4 (CTLA-4) [4], B and T-lymphocyte attenuator (BTLA) [5], T cell immunoglobulin and ITIM domain (TIGIT) [6], and programmed death-1 (PD-1). ICOS, PD-1 and CTLA-4 are not constitutively expressed on resting T cells but are rapidly upregulated after activation [4, 7]. ICOS is expressed on activated CD4 and CD8 T cells; it regulate various T helper cell subsets by promoting or inhibiting Th1 and Th2 immune responses [8]. CTLA-4 decreases T cell receptor (TCR) signalling through competition with the co-stimulatory molecule CD28 for the ligands B7-1 (CD80) and B7-2 (CD86), for which CTLA-4 has higher avidity and affinity. CTLA-4 is also highly expressed on TREGs and is essential for their suppressive function and the maintenance of peripheral tolerance [9, 10]. BTLA (CD272) is mainly expressed on B cells, T cells, and all mature lymphoid cells; it inhibits T cell responses and cytokine production [11]. BTLA expression is low on naive CD4$^+$T cells [12] but increases after activation [13]. However, BTLA expression remains low on TREGs [12]. TIGIT is specifically expressed in immune cells, where it acts as a co-inhibitory receptor in parallel to the CD28/CTLA-4 pathway [14]. In both humans and mice, TIGIT is highly expressed on a subset of natural TREG and marks an activated TREG phenotype [15]. Compared to TIGIT$^-$ TREG, TIGIT$^+$ TREG demonstrated to be superior in suppressing T cells. The other important member of this family (PD-1) delivers inhibitory signals that regulate the balance between cell activation, tolerance, and immune disease [16]. PD-1's inhibitory functions have been best characterized in CD4$^+$ and CD8$^+$ TEFFs, and it has been demonstrated that PD-1 signals are also important for induced T regulator cell development [17, 18]. In a mouse model, PD-1 expression on TREGs was upregulated upon activation [19], and PD-1 blockade appears to lower the TREGs' suppressor activity *in vivo* [20]. PD-L1 (also known as B7 homolog 1 (B7-H1) or cluster of differentiation 274 (CD274) and PD-L2 (B7-DC, CD273) are PD-1 ligands from the B7 family. PD-L1 is expressed on hematopoietic cells (such as resting B cells, myeloid cells and dendritic cells (DCs)) and is upregulated after activation [21]. Indeed, induction of PD-L1 expression has been described on CD3/CD28 activated polyclonal CD4+CD8 + human T cells and also on T cells derived from tumor [22]. This expression has also been described in an unique activated γδ T cells population in pancreatic ductal adenocarcinoma (PDA) [23]. However, there is no evidence to date of PD-L1 expression on human naïve CD4$^+$ T cells. Indeed, high levels of PD-L1 mRNA (but not PD-L1 protein) have been detected on activated naïve CD4$^+$ or CD8$^+$ T cells but not on other T cell populations (including non-activated CD4$^+$ or CD8$^+$ T cells, follicular T helper cells, T helper 1 cells, Th2 cells, Th17 cells, etc.) (https://dice-database.org). PD-L2 expression is more restricted and is induced on DCs, macrophages, and bone marrow–derived mast cells. This PD-L2 expression cannot compensate for the lack of PD-L1 in regulating T cell responses [24]. PD-L1 interacts with several receptors (PD-1 and CD80) and can induce various signals by either direct interaction or binding competition with CD28 or CTLA-4 (both of which bind to CD80). The second family of co-signalling molecules is the tumour necrosis factor receptor (TNFR) superfamily (TNFRSF), which includes CD27, glucocorticoid-induced TNFR family-related (GITR), and 4-1BB/CD137 [25]. The TNFRSF members are expressed on various cell types (including T cells) and are involved in the regulation of TCR signalling [26–28]. The majority of naive peripheral CD4$^+$ and CD8$^+$ T cells express CD27, which is upregulated upon TCR stimulation. 4-1BB is also expressed on TREGs [29] and GITR is important for the development and maturation of TREGs [25]. However, other receptors might be involved in the regulation of T cell activation, such as T cell

immunoglobulin and mucin domain-containing protein 3 (TIM3) and lymphocyte activation gene 3 (LAG3) [30]. TIM3 (a member of the TIM family of immune regulatory receptors) is expressed predominantly on Th1 cells but also on Th17 cells. It appears to be a co-inhibitory molecule for T cell function [28] but is weakly expressed by conventional CD4[+] T cells and by TREGs in the peripheral blood [31]. LAG3 (CD223) bears structural homology to CD4 and is upregulated on activated CD4[+] and CD8[+] T cells [32]. It is also expressed by activated natural TREGs and induced CD4[+]FoxP3[+] TREGs, where its expression levels are higher than those observed on activated CD4[+] TEFFs [33].

In the present study, we used an *in vitro* cell activation assay (developed previously for studies of autoimmune diseases) to investigate the surface expression of these receptors on TEFFs from healthy controls; the TEFFs were activated by a combination of DCs and staphylococcal enterotoxin E (SEE) and were regulated by TREGs [34, 35]. Our results show that a change in the surface receptor profile on TEFFs is correlated with the regulation of the TEFF proliferation by TREGs.

# Materials and methods

## Cell isolation

Purified peripheral blood mononuclear cells (PBMCs) from 12 healthy adult controls (8 females and 4 males; mean ± standard deviation age: 34 ± 15 (range: 18–58); French Blood Transfusion Service, Paris, France) were prepared by density gradient centrifugation on Lymphoprep (Abcyss SA) and tested in independent experiments. The donors' TEFFs, TREGs and DCs were sorted using flow cytometry, as described elsewhere [34]. All three cell preparations were at least 90% pure.

## Antibodies and reagents

For the sorting experiments, FITC-coupled anti-CD4 and PE-coupled anti-CD25 were purchased from BD Biosciences (MountainView, CA, USA), APC anti-CD45RA was purchased from BioLegend (San Diego, CA, USA), PE-Cy5 anti-CD11c was purchased from Beckman Coulter (Fullerton, CA, USA), BV650 anti-CD3, PE-Cy7 anti-CD127 and BV510 anti-CD16 were purchased from Sony Biotechnology (San Jose, CA, USA), and Violet Blue anti-CD14 was purchased from Miltenyi Biotec (Bergisch Gladbach, Germany). For the proliferation assays, SEE was purchased from Toxin Technology Inc (Sarasota, FL, USA), CellTrace™ carboxyfluorescein succinimidyl ester (CFSE) was purchased from Molecular Probes (Eugene, OR, USA) and CellTrace™ Violet (CTV) was purchased from Invitrogen (Carlsbad, CA, USA). For phenotyping, PE-anti-PD-L1, PE-anti-BTLA, PE-anti-LAG3, PE-anti-TIM3, APC-anti-PD-1, and APC-anti-4-1BB were purchased from BioLegend (San Diego, CA, USA), APC anti-ICOS was purchased from Invitrogen (Carlsbad, CA, USA) and APC anti-TIGIT was purchased from Sony Biotechnology (San Jose, CA, USA). PE-anti-PD-L2 and anti-CTLA-4 were purchased from ebiosciences (San Diego, CA, USA).

## Purification of T cells from PBMCs and co-culture with sorted DCs

PBMCs were incubated for 30 min at 4˚C with specific, labelled monoclonal antibodies, washed and then sorted using a cytometer (ARIA II, BD Biosciences). Naive TEFFs were defined as CD3[+]CD4[high]CD25[low]CD127[high]CD45RA[+] T cells, TREGs were defined as CD3[+]CD4[high]CD25[+]CD127[low] T cells, and DCs were defined as CD3[-]CD4[low]CD11c[+]CD14[-]CD16[-] cells. To assess cell proliferation, sorted naive TEFFs and TREGs were washed and then stained with CTV and CFSE, respectively. Next, the cells were

washed and incubated with DCs ($8x10^3$ TEFFs and $2x10^3$ DCs, giving a ratio of 4:1). TREGs were added to give a TEFF:TREG ratio of 1: 0.5 or 1: 0.125. SEE (0.2 ng/ml) was then added. The cell proliferation assay was performed with autologous and heterologous samples in Pan-serin medium (Dutscher, Brumath, France) supplemented with 5% human AB serum in 96-well plates. After 4, 5 or 7 days of culture, the various conjugated antibodies were added to the culture for 20 min in the dark at room temperature. The percentage and phenotype of the two proliferating T cell subpopulations were measured with a MACSquant system (Miltenyi), and the data were analyzed with FlowJo software. The phenotyping results were expressed as the geometric mean and normalized against the mode. Statistical analyses were performed using GraphPad Prism software (version 6, GraphPad Software, Inc., La Jolla, CA). A non-parametric Mann-Whitney (M-W) test was used to compare the data for the various markers.

All study participants provided written informed consent. The study was performed in accordance with the 1975 Declaration of Helsinki and subsequent revisions, and was approved by the local institutional review board (CPP Ile de France II Paris, France), the French Advisory Committee on Data Processing in Medical Research (Paris, France).

## Results

### PD-L1 expression on human naive CD4⁺ TEFFs co-cultured with DCs

We first studied the phenotype of naive human CD4⁺ TEFFs at different time points after activation with DCs and SEE in a previously described *in vitro* assay [34]. The highest TEFF proliferation index was observed on day 5 (D5) (Fig 1A). We found that this time point corresponds to a peak in the expression of the various different receptors (such as PD-1 and ICOS) known to have a regulatory role in T cell activation (Fig 1B). Surprisingly, we also observed significant PD-L1 expression on activated TEFFs on D5 (Fig 1B). Furthermore, a lower proliferation index on D7 was correlated with significantly lower PD-L1 expression (Fig 1B). In contrast, the other PD-1 ligand (PD-L2) was weakly expressed (S1 Fig). We also observed that on TEFFs, (i) BTLA was strongly expressed, (ii) TIM3, 4-1BB, CTLA-4 and LAG3 were moderately expressed, and (iii) GITR was not expressed (S1 Fig). CTLA-4 expression was most easily detected on D5 (S2A Fig). By testing cells from 10 controls several times, we observed that the activated TEFFs' expression of PD-1, ICOS and PD-L1 varied from one individual to another (Fig 1C). This finding indicates that human peripheral blood naïve CD4⁺ T cells stimulated with DCs and SEE express PD-L1. Moreover, the highest level of PD-L1 expression was observed on D5 and was thus correlated with the highest proliferation index.

### PD-L1 expression on activated naive TEFFs is correlated with DC efficacy

We found two groups of control cells with regard to the TEFF proliferation index on D5 (Fig 2A): one group had a high proliferation index, and the other group had a lower proliferation index. Interestingly, a comparison of PD-L1 expression in the two groups showed that PD-L1 was most strongly expressed on activated TEFFs with the highest proliferation index (Fig 2B). In contrast the PD-1 and ICOS expressions were similar in both groups of samples (Fig 2B). This result suggests that the level of PD-L1 expression on TEFFs is linked to the proliferative response induced by DC and SEE.

### PD-L1 expression on activated naive TEFFs is regulated by TREGs

We reported previously that TEFF proliferation can be significantly inhibited by TREGs when the TEFF:TREG ratio is 1:0.5 [34]. Here, we confirmed that TREGs can even inhibit TEFFs with a high proliferative index (Fig 3A and S3A Fig). We therefore looked at whether the

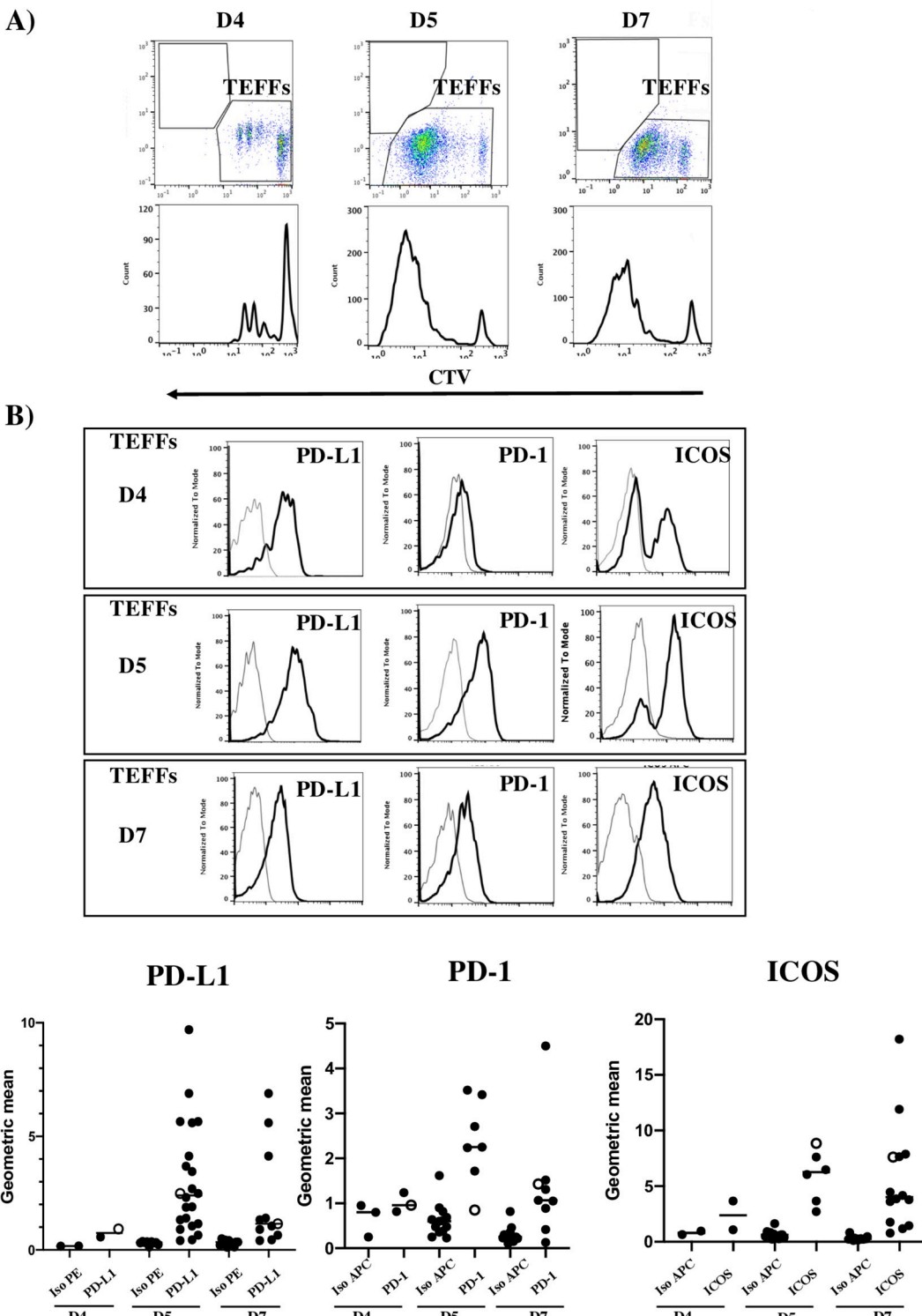

**Fig 1. PD-L1 expression on activated naive TEFFs.** A) Representative flow cytometry dot plots and histograms for the proliferation of CTV-stained TEFFs on D4, D5 and D7 of the co-culture. B) Upper panels: representative histograms of PD-L1, PD-1 and ICOS on the gated CTV-positive cells at the different culture time points. Lower panels: graphs of the geometric mean fluorescence intensity of the respective markers in independent experiments (n = 2 to 10). For each receptor, the specific conjugated isotype control (grey solid lines) is shown (Iso PE or Iso APC). Each black dot represents an experiment, and the open circles represent data from the same experiment. Bars represent the means of all experiments for each proteins.

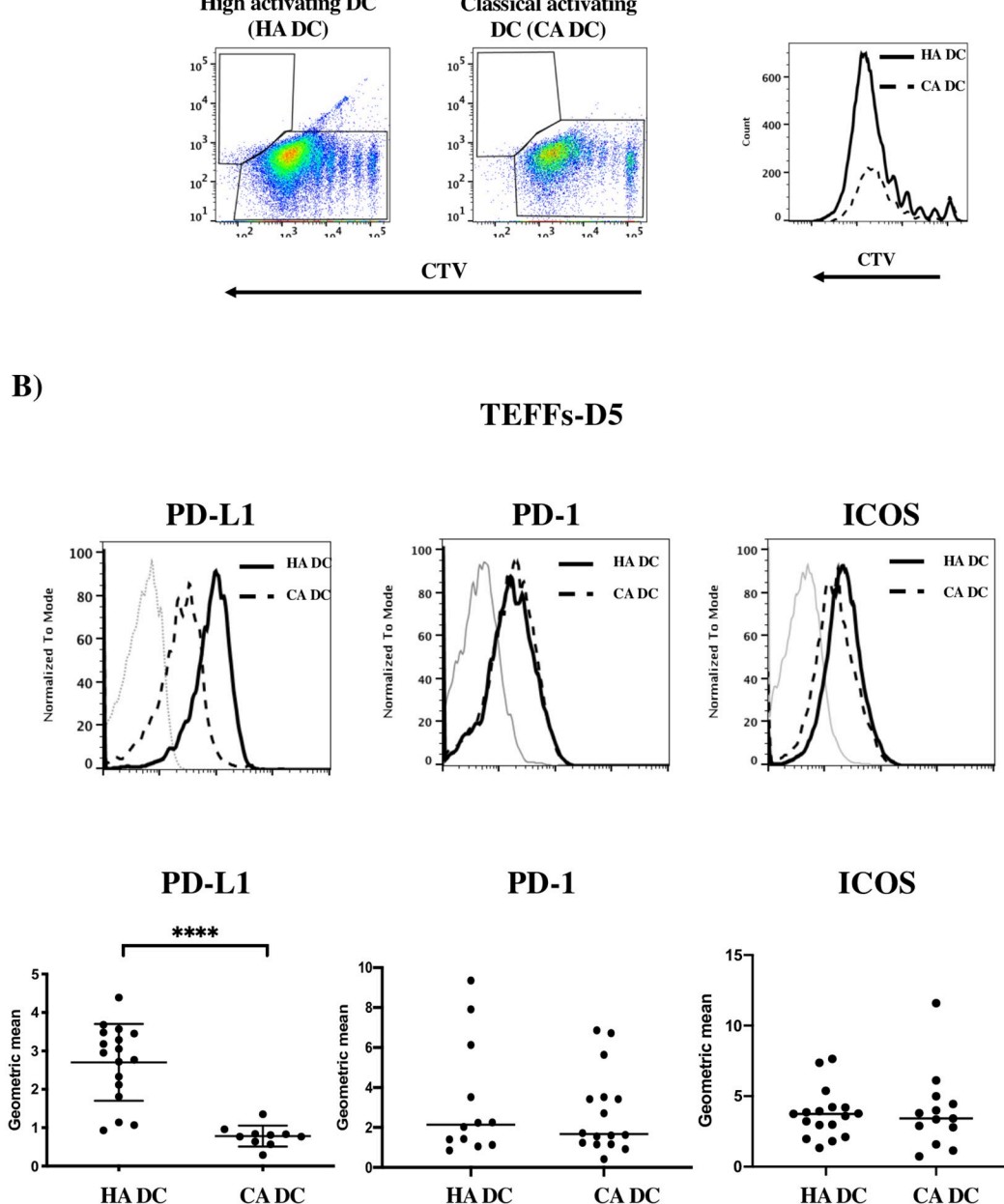

**Fig 2. PD-L1 expression on activated naive TEFFs is correlated with DC efficacy.** A) Representative flow cytometry dot plots and overlay histogram for the proliferation of CTV-stained TEFFs on D5 of the co-culture, for individual samples with a high proliferative index with strongly activating DCs (HA DC) (solid black line) or a low proliferative index with conventional activating DCs (CA DC) (dotted black line). B) Upper panel: Representative overlay histograms of PD-L1, PD-1 and ICOS expression on CTV-stained TEFFs from the group with a high proliferative index (solid black lines) and the group with a low proliferative index (dashed black line). The isotype control (Iso) is shown for each protein (solid grey lines). Lower panel: graphs of each respective marker's geometric mean fluorescence intensity in independent experiments (n = 14 for strongly activating (SA) DCs, n = 12 for conventional activating (CA) DCs). **** p≤0.0001 in an M-W test. Bars represent the means of all experiments for each protein. Each black dot represents an experiment.

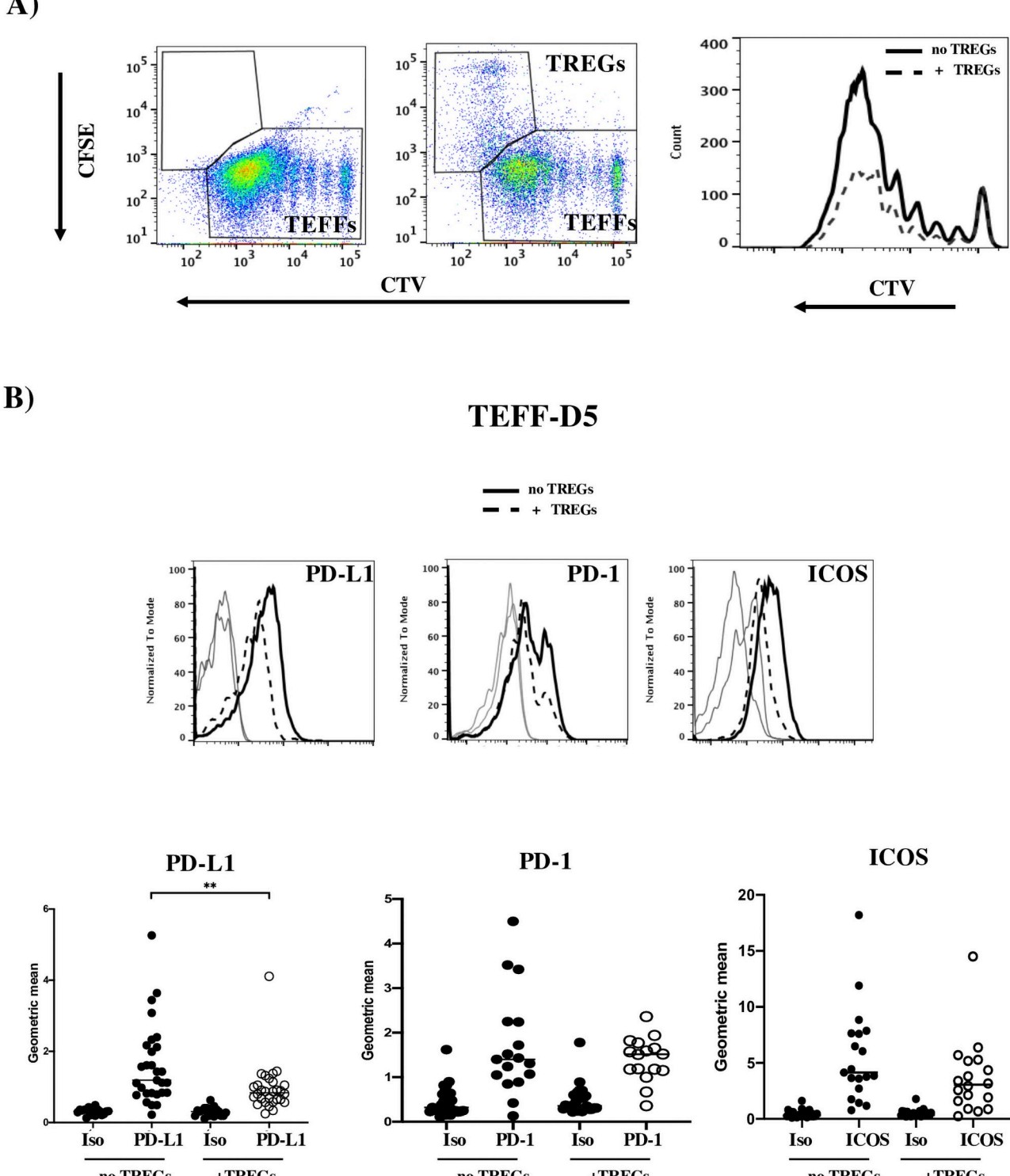

**Fig 3. PD-L1 expression on activated naive TEFFs is regulated by TREGs.** A) Representative flow cytometry dot plots and histograms for the proliferation of CTV-stained TEFFs on D5 of co-culture, in the absence of TREGs (left dot plot and the solid black line on the histogram) or in the presence of CFSE-stained TREGs with a TEFF:TREG ratio of 1:0.5 (right dot plot and the dashed black line on the histogram). B) Representative overlay histograms of different surface receptors in the absence (solid black lines) or presence of TREGs (dashed black line). The isotype control (Iso) is shown for each protein (solid grey lines). Lower panel: graphs of each respective marker's geometric mean fluorescence intensity in independent experiments (n = 8 to 25). ** p≤0.01 in an M-W test. Bars represent the means of all experiments for each proteins.

PD-L1 expression on activated TEFFs was affected by co-culture with TREGs. Indeed, expression levels of PD-L1 (Fig 3B) and CD27 (S3B Fig) on TEFFs were significantly lower in the presence of TREGs. Expression levels of PD-1 and ICOS were slightly but not significantly lower (Fig 3B). In contrast, the expression levels of PD-L2, BTLA, LAG3 TIM3 and CTLA4 did not depend on the presence or absence of TREGs (S2B and S3B Figs). At a high TEFF: TREG ratio (1:0.125), there was no inhibition of proliferation (Fig 4A). Under these conditions, the PD-L1 expression on the TEFFs was similar to that observed in the absence of TREGs (Fig 4B). However, the significant inhibition of TEFF proliferation (Fig 4A) observed with a TEFF:TREG ratio of 1:0.5 was correlated with significantly lower PD-L1 expression on TEFFs (Fig 4B). This result confirms the link between PD-L1 expression on activated TEFFs and their proliferation index.

## Expression of co-signalling receptors on activated TREGs

By using distinct cell tracers, we could simultaneously assess the expression of receptors on CFSE-stained TREGs and CTV-stained TEFFs co-cultured with DCs and SEE on D4, D5 and D7 (Fig 5). We observed that the TREG-induced inhibition of proliferation was greatest on D5 of the co-culture (Fig 5A and S5A Fig), which was again the best time to detect PD-L1 expression on TREGs (Fig 5B). Concerning the expression of other receptors on D5, TREGs expressed PD-1, ICOS, and CTLA-4 strongly and PD-L2, BTLA, LAG3 and TIM3 weakly at all timepoints (Fig 5B and S4 Fig). As described for PD-L1 (Fig 5B), we observed that CTL-4 expression decreased significantly on D7 of co-culture (S2C Fig). To confirm the absence of possible contamination between gated TEFFs and TREGs, we also assessed TIGIT expression because it has been reported previously that this protein is mainly expressed on suppressor TREGs [36]. Marked TIGIT expression by TREGs was observed on D5 of co-culture (S4 Fig). In contrast, TIGIT expression was never detected on TEFFs co-cultured with DCs in the absence or in the presence of TREGs—showing that the absence of TIGIT expression on TEFFs was not linked to inhibition by TREGs (S5B Fig). Overall, these results showed for the first time that PD-L1 can also be expressed on human peripheral CD4$^+$ TREGs co-cultured with CD4$^+$ TEFFs activated by DCs and SEE. This observation suggests that the three types of cell interact in novel ways to regulate proliferation.

## Discussion

In the present study, we assessed the surface expression of several receptors known to be involved in the regulation of T cell activation. We observed that upon activation with DCs and SEE, naive CD4$^+$ TEFFs expressed several receptors, such as BTLA, LAG3, 4-1BB and TIM3. In contrast, the TEFFs did not express GITR—suggesting that the expression of this receptor requires longer activation. The expression of PD-1 and ICOS—two key receptors involved in the regulation of T cell activation—was also detected on TEFFs on D5 of co-culture. Interestingly, we observed significant expression of one of PD-1's ligands (PD-L1) on TEFFs under these conditions. The receptor PD-1 is reportedly expressed on antigen presenting cells, such as macrophages and DCs. Our results are consistent with a report on a mouse model in which TCR activation increased the surface expression of both PD-L1 and PD-1 on T cells from the spleen and lymph nodes [37]. In contrast, we observed that PD-L2 was not detected on activated TEFFs. Unfortunately, we could not compare the PD-L1 expression on T cells vs. DCs because the latter died after three days of co-culture (data not shown). PD-L1 expression on T cells was correlated strongly with their proliferative capacity. This might result from a DC-mediated signal, through either direct contact between T cells and DCs during the first three

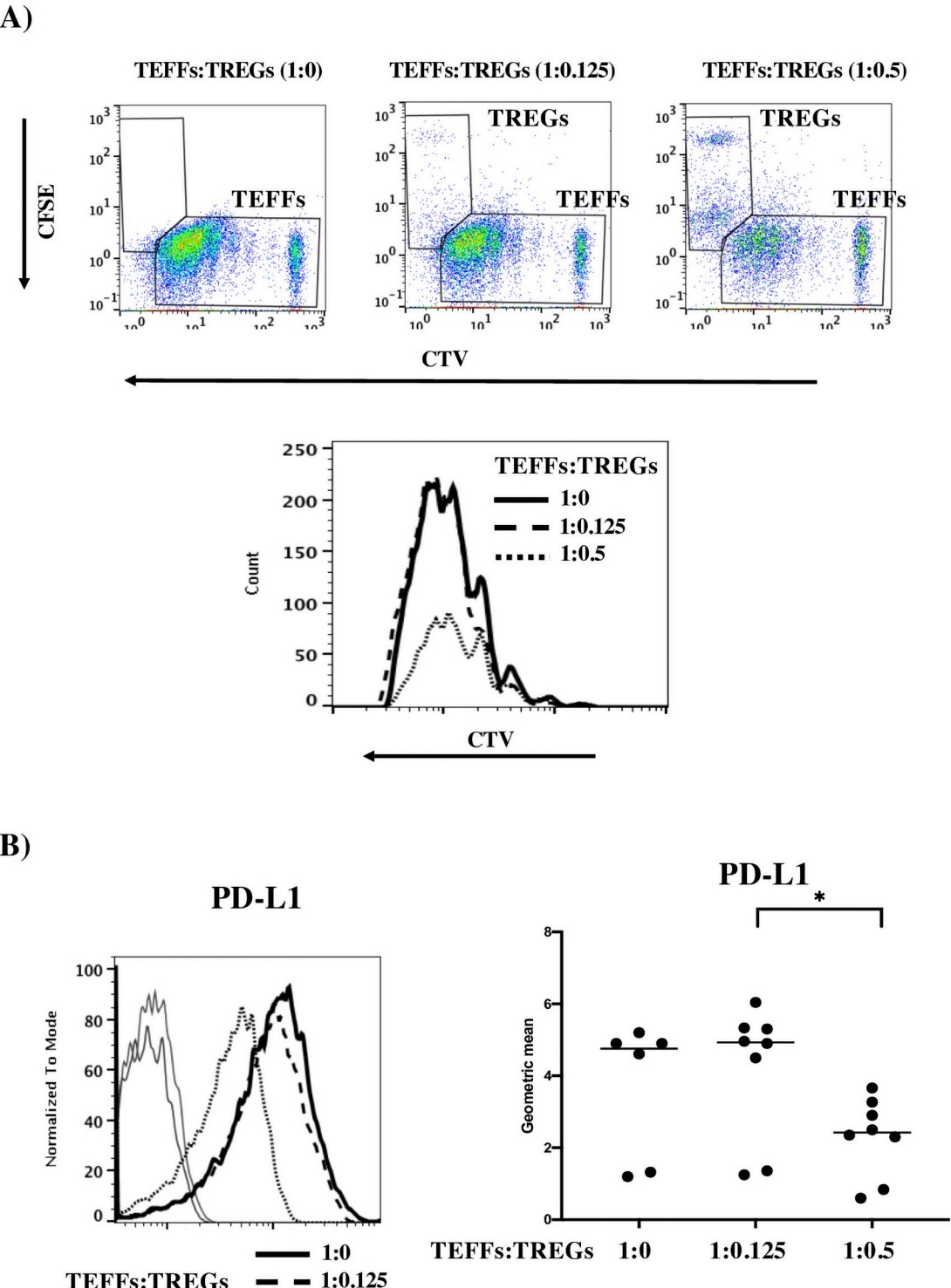

**Fig 4. PD-L1 expression on activated TEFFs is dependent on the TEFF:TREG ratio.** A) Representative flow cytometry dot plots for the proliferation of CTV-stained TEFFs on D5 of co-culture, with different TEFF: CFSE-stained TREG ratios (1:0 in the left panel, 1:0.125 on the middle panel, and 1:0.5 on the right panel). Lower panel: Representative overlay histograms for the proliferation of CTV-stained TEFFs with different TEFF:TREG ratios: 1:0 (solid black lines), 1:0.125 (dashed black lines), or 1:0.5 (dotted lines). B) Left panel: a representative overlay histogram of PD-L1 expression on activated, CTV-stained TEFFs with a TEFF:TREG ratio of 1:0 (solid black line), 1:0.125 (dashed line) or 1:0.5 (dotted line). Right panel: a graph of PD-L1's geometric

mean fluorescence intensity on CTV-stained TEFFs in 6 to 8 independent experiments. Bars represent the means of all experiments for each proteins. The isotype control (Iso) is shown for each protein (solid grey lines). * p≤0.05 in an M-W test.

days of culture or paracrine or autocrine production of one or more cytokines. Indeed, PD-L1 expression was moderate on D4 of co-culture and peaked on D5.

We also observed that PD-L1 expression (but not ICOS and PD-1 expression) was correlated with a higher proliferative index in TEFFs. By performing several assays on the same samples, we observed two groups of controls (one with a high proliferative index on D5 and one with a low index D5) and thus highlighted interindividual variability. Strikingly, the TEFFs with the highest proliferative index expressed PD-L1 most strongly. In contrast, PD-1 and ICOS expression levels were not correlated with the TEFFs' proliferation index.

PD-L1 surface expression on activated CD45RA+ TEFFs was unexpected although already described on purified CD3+CD4+ T cells only after activation with anti-CD3/CD28/IgG-coated beads but not with PD-L1-Ig [38]. In addition, PD-1 expression was well induced on these purified T cells activated in both conditions. However, as mentioned above, PD-L1 mRNA has been detected on activated naive $CD4^+$ or $CD8^+$ T cells but not on non-activated T cell populations (https://dice-database.org). Our present results are in line with these findings. Furthermore, we showed previously that TREGs inhibit SEE/DC-induced T cell proliferation [34]. In mechanistic terms, this inhibition might involve three-way contact between TREGs, TEFFs, and targeting DCs via interactions with inhibitory receptors [39]. We therefore investigated the regulation of receptor expression on activated T cells in the presence of TREGs. We observed lower expression of PD-L1 and CD27 on TEFFs co-cultured with DCs and TREGs when the TEFF:TREG ratio efficiently inhibited TEFF proliferation. Expression levels of the other receptors were not affected by the presence of TREGs.

Importantly, we also observed PD-L1 expression on the TREGs in the co-culture. Significant PD-1 and PD-L1 expression on DC-activated TREGs (suggesting that the PD1/PD-L1 axis modulates both cells) has been reported previously [40]. Amarnath et al. suggested that PD-L1-expressing TREGs directly increased the expression of PD-L1 on DCs. We could not study the impact of PD-L1-expressing TREGs on DCs since the latter died after three days of culture, at which time PD-L1 was not expressed on the TREGs. This prevented us from determining whether the PD-L1 expression on TREGs was induced by DCs and/or activated TEFFs.

With regard to other receptors, we observed that ICOS, PD-1 and CD27 are expressed at the same time on TREGs. In contrast, PD-L2, BTLA, LAG3, TIM3 and GITR (data not shown: similar to TIM3) are poorly expressed on TREGs. Interestingly, other researchers have reported that GITR is expressed on TREGs [39] but this discrepancy might be due to a difference in activation (SEE in our study and transforming growth factor beta in the study by Lohr et al.). We also assessed the expression of CTLA-4, a key inhibitor involved in the TREGs' suppressor function. We found that activated TEFFs expressed CTLA-4 and that (in contrast to PD-L1 expression) this expression did not decrease in the presence of TREGs. Lastly, we observed CTLA-4 expression on activated TREGs, as reported previously for various disease and tumour models [41, 42].

Taken as a whole, our results show that naive $CD4^+$ T cell proliferation stimulated by DCs/SEE and regulated by TREGs is correlated with a specific change in PD-L1 surface expression. The PD-L1 expression on TEFFs and TREGs suggests a complex interaction possibly involved in the regulation of T cells. The fact that all three cell partners express PD-L1 suggests cis- and/or trans-interactions with either the classical PD-L1 receptor (PD-1) or another PD-L1 receptor with different affinity/avidity. We did not seek to determine the consequences of these

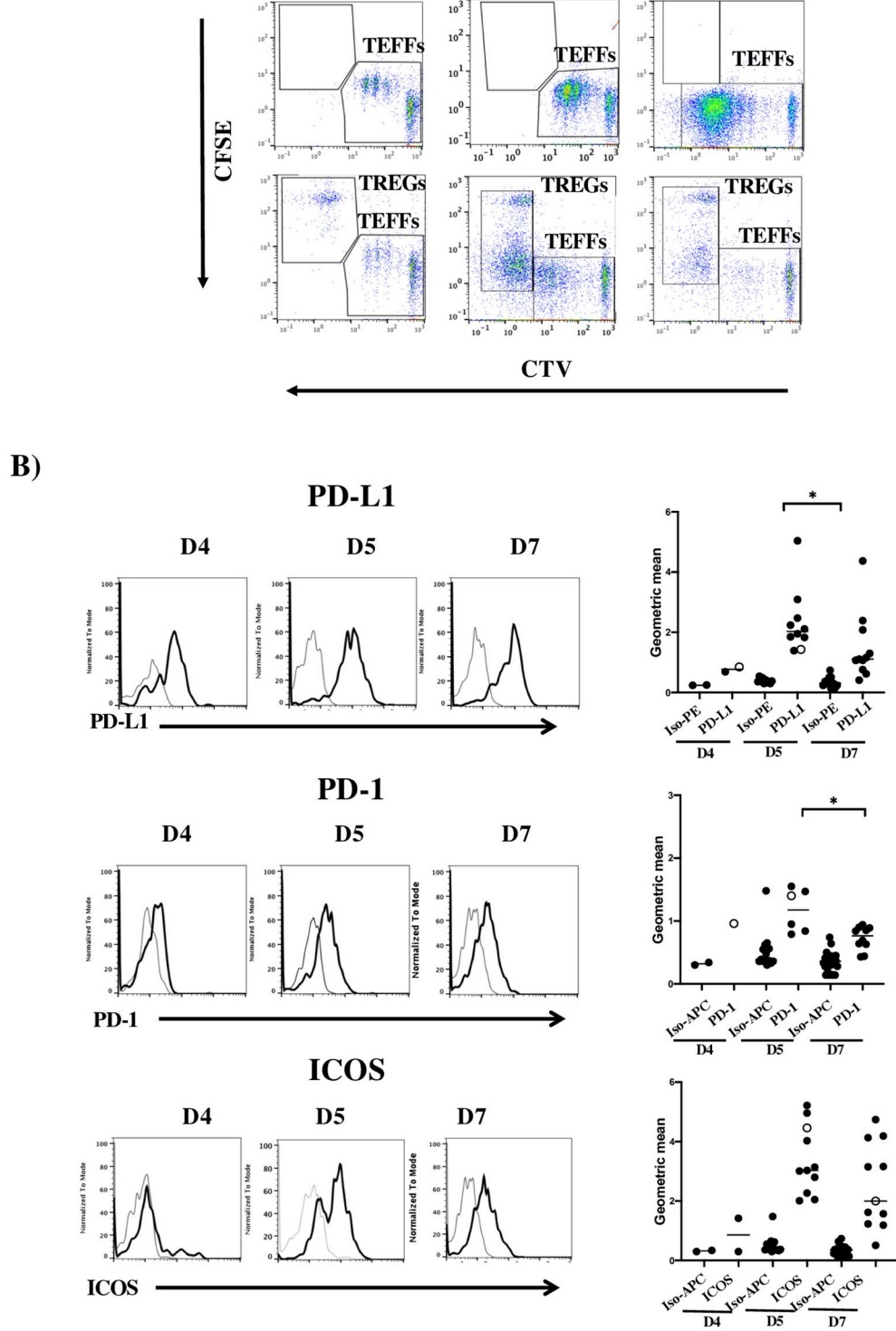

**Fig 5. Surface expression of PD-L1, PD-1 and ICOS on activated TREGs.** A) Representative flow cytometry dot plots for the proliferation of CTV-stained TEFFs on D4, D5 and D7 of co-culture in the absence or the presence of CFSE-stained TREGs. The TEFF:TREG ratio was 1:0 in the top plot and 1:0.5 in the bottom plot. B) Left panel: representative histograms of receptor expression (solid black lines) on the gated, CFSE-stained TREGs on D4, D5 and D7 of co-culture. Right panel: graphs of each marker's geometric mean fluorescence intensity at the different co-culture time points for CFSE-stained TREGs in independent experiments (n = 2 to 12, depending on the marker and

the time course studied). The isotype control (Iso) is shown for each marker (grey lines). $^*$ p$\leq$0.05 in an M-W test. Bars represent the means of all experiments for each proteins. Each black dot represents an experiment, and the open circle represents the same experiment.

interactions, which might serve to limit TCR signalling after activation; indeed, the inhibition of proliferation was correlated with a decrease in PD-L1 expression on TEFFs. Therefore, PD-L1 expression on TEFFs might be an interesting biomarker of autoimmune disease; it might help physicians to identify and thus target the patient's most strongly proliferative T cells.

## Supporting information

**S1 Fig. Expression of co-signalling receptors on activated naive TEFFs.** Upper panel: representative histograms of surface receptors on gated, CTV-stained cells on D5 of proliferation assay. For each protein, the specific conjugated isotype control (grey solid line) is shown (Iso-PE- or Iso-APC). Lower panel: graphs of the geometric mean fluorescence intensity of the respective markers in independent experiments (n = 2 to 9). Bars represent the means of all experiments for each proteins.
(TIF)

**S2 Fig. Expression of membrane CTLA-4 on activated T cells.** A) Left panel: representative histograms of CTLA-4 expression on gated CTV-stained TEFFs co-cultured with DCs-SEE for 5 days (D5) or 7 days (D7). Right panel: graphs of the geometric mean fluorescence intensity of CTLA-4 expression in independent experiments (n = 6 on D5 and D7). The isotype control (Iso) is shown. B) Left panel: representative histograms of CTLA-4 expression on gated, CTV-stained TEFFs co-cultured with DCs-SEE in the absence (no TREGs) or presence of TREGs (+TREGs) for 5 days (D5). The TEFF:TREG ratio was 2:1. Right panel: graphs of the geometric mean fluorescence of CTLA-4 expression in independent experiments (n = 13 on D5). C) Left panel: representative histograms of CTLA-4 expression on gated, CFSE-stained TREGs co-cultured for 5 days (D5) or 7 days (D7). Right panel: graphs of the geometric mean fluorescence intensity of CTLA-4 expression in independent experiments (n = 15 on D5, n = 6 on D7). $^*$ p$\leq$0.05 in a M-W test, $^{**}$ p$\leq$0.01 in in an M-W test. Bars represent the means of all experiments for each proteins.
(TIF)

**S3 Fig. Co-signalling receptor expression on activated naive TEFFs in the presence of TREGs.** A) Representative flow cytometry dot plots and histograms of the proliferation of CTV-stained TEFFs on D5 of co-culture, in the absence (left dot plot and the solid black line on the histogram) or presence of CFSE-stained TREGs and a TEFF:TREG ratio of 2:1 (right dot-plot and dashed black line on the histogram). B) Representative overlay histograms of different surface receptors in the absence (solid black lines) or presence of TREGs (dashed black line). The isotype control (Iso) is shown for each protein (solid grey lines). Lower panel: graphs of the geometric mean fluorescence intensity of each of the markers, in independent experiments (n = 3 to 8). Bars represent the means of all experiments for each proteins.
(TIF)

**S4 Fig. Co-signalling receptor expression on activated TREGs.** Upper panel: representative histograms of different receptors expressions (solid black lines) on the gated CFSE-stained TREGs on D5 of the co-culture. Lower panel: graphs of each marker's geometric mean intensity on D5 (n = 2 to 12, depending on the marker and the time course studied). The isotype

control (Iso) is shown for each protein (grey lines). Bars represent the means of all experiments for each proteins.
(TIF)

**S5 Fig. TIGIT expression on activated T cells.** A) Representative flow cytometry dot plots for cell proliferation on D5 of the co-culture. CTV-stained TEFFs were co-cultured with DCs-SEE in the absence (TEFFa) or the presence of CFSE-stained TREGs (TEFFb) with a TEFF:TREG ratio of 2:1. B) Representative histograms of TIGIT expression (solid black lines) on gated CTV-stained TEFFs in the absence of TREGs (TEFFa) or in the presence of TREGs (TEFFb) and on gated CFSE-stained TREG (TREGs). The TEFF:TREG ratio was 2:1. The isotype control (Iso-APC) is shown for each population (grey lines). Lower panel: graphs of the geometric mean fluorescence intensity of TIGIT expression on gated, CTV-stained TEFFs (TEFFa and TEFFb) and on gated, CFSE-stained TREGs in independent experiments (n = 5 to 30, depending on the sub-population studied). **** $p \leq 0.0001$ in an M-W test. Bars represent the means of all experiments for each proteins.
(TIF)

## Acknowledgments

We thank to O. Pelle and J. Megret from the SFR Necker Cytometry facility, INSERM 1163, INSERM US24 and CNRS UMS3633 for the cell sorting on the ARIA system.

## Author Contributions

**Conceptualization:** Fabienne Mazerolles.

**Data curation:** Fabienne Mazerolles, Frédéric Rieux-Laucat.

**Formal analysis:** Fabienne Mazerolles.

**Funding acquisition:** Frédéric Rieux-Laucat.

**Investigation:** Fabienne Mazerolles.

**Methodology:** Fabienne Mazerolles.

**Supervision:** Frédéric Rieux-Laucat.

**Validation:** Fabienne Mazerolles.

**Writing – original draft:** Fabienne Mazerolles, Frédéric Rieux-Laucat.

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
