## [Decision Letter · Decision Letter 0]

27 Sep 2021

PONE-D-21-22116PD-L1 IS EXPRESSED ON HUMAN ACTIVATED NAIVE EFFECTOR CD4+ T CELLS. REGULATION BY DENDRITIC CELLS AND REGULATORY CD4+ T CELLSPLOS ONE

Dear Dr. Mazerolles,

Thank you for submitting your manuscript to PLOS ONE. After careful consideration, we feel that it has merit but does not fully meet PLOS ONE’s publication criteria as it currently stands. Therefore, we invite you to submit a revised version of the manuscript that addresses the points raised during the review process.

We consider that the manuscript lacks precision in various aspects, and the data presented need to be  completed and strengthened. We invite you to submit a revised version of the manuscript that addresses all the points raised during the review (see below Reviewer's comments).

We look forward to receiving your revised manuscript.

Kind regards,

Prof. Pierre Bobé

Academic Editor

PLOS ONE

Journal Requirements:

The study was funded by the Institut National de la Santé et de la Recherche Médicale (INSERM), a grant from the French government (managed by French National Research Agency (Agence National de la Recherche) as part of the “Investment for the Future” program (Institut Hospitalo-Universitaire Imagine, grant ANR-10-IAHU-01, Recherche Hospitalo-Universitaire, grant ANR-18-RHUS-0010)), and other grants from the Agence National de la Recherche (ANR-14-CE14-0026-01 “Lumugene”; ANR-18-CE17-0001 “Action”), the Fondation pour la Recherche Médicale (Equipe FRM EQU202103012670 ), the Ligue Contre le Cancer – Comité de Paris, Fondation ARC pour la recherche sur le CANCER, and the Centre de Référence Déficits Immunitaires Héréditaires (CEREDIH).

The author received no specific funding for this work

Reviewers' comments:

Reviewer's Responses to Questions

**Comments to the Author**

1. Is the manuscript technically sound, and do the data support the conclusions?

Reviewer #1: Partly

2. Has the statistical analysis been performed appropriately and rigorously? 

Reviewer #1: Yes

3. Have the authors made all data underlying the findings in their manuscript fully available?

Reviewer #1: Yes

4. Is the manuscript presented in an intelligible fashion and written in standard English?

Reviewer #1: Yes

5. Review Comments to the Author

Reviewer #1: In the this study the authors present their observation that PD-L1 is expressed in activated human CD4+ T cells. They assess correlation expression levels of PD-L1 and other positive and negative costimulatory receptors such as ICOS, PD-1, CTLA4, TIGIT in T effector and Treg cells during culture of naïve CD4+ T cells with SEE superantigen and dendritic cells.

They found that activated T effector cells expressed BTLA, ICOS, PD-1 and PD-L1, whereas Treg cells expressed TIGIT, ICOS and CD27. They also observed that expression of PD-L1 in T effector cells positively correlated with their proliferation during exposure to SEE in the presence of DC. Titration experiments of Treg addition in the culture showed that suppression of T effector cell activation by the addition of Treg resulted in diminished expression of PD-L1. Based on these findings, the authors proposed that PD-L1 expression on T effector cells might serve as a marker of their activation and might be a useful biomarker in conditions of aberrant T cell activation such as in autoimmune diseases.

The results are interesting but not surprising based on our knowledge about the expression of these surface receptors in T cell subsets, including T effector cells and Treg.

Specific points:

1) Several previous studies have shown that PD-L1 is expressed in activated T cells (e.g. Bardhan et al. Sci Rep 2019; 9: 17252; Diskin et al. Nat Immunology 2020; 21: 442). Thus, this observation is not made for the first time as stated by the authors.

2) The authors correlated the expression of PD-L1 with the proliferation state of T effector cells. It is important to examine cytokine production by T cells under the same conditions.

3) The studies as they stand are only correlative and the role of PD-L1 in regulating the function of T effector and Treg cells has not been examined. This aspect should be addressed by using appropriate antibodies targeting each of these surface receptors.

4) All the experiments have been performed using SEE. It is well established that superantigens overcome the standard signals medicated by TCR and accessory molecules, including co-inhibitory and co-stimulatory receptors. For this reason, it would be more appropriate to assess the proposed role of these surface molecules of their interest in an antigen-specific system.

6. PLOS authors have the option to publish the peer review history of their article (what does this mean?). If published, this will include your full peer review and any attached files.

Reviewer #1: No

---

## [Author Response · Author response to Decision Letter 0]

26 Oct 2021

RESPONSE TO REVIEWER

Journal requirements. 

We have now ensured that our manuscript meets the style requirements of PLOS ONE

We have checked the grant numbers for the awards .

The study was funded by the Institut National de la Santé et de la Recherche Médicale (INSERM), a grant from the French government (managed by French National Research Agency (Agence National de la Recherche) as part of the “Investment for the Future” program (Institut Hospitalo-Universitaire Imagine, grant ANR-10-IAHU-01, Recherche Hospitalo-Universitaire, grant ANR-18-RHUS-0010)), and other grants from the Agence National de la Recherche (ANR-14-CE14-0026-01 “Lumugene”; ANR-18-CE17-0001 “Action”), the Fondation pour la Recherche Médicale (Equipe FRM EQU202103012670 ), the Ligue Contre le Cancer – Comité de Paris, Fondation ARC pour la recherche sur le CANCER, and the Centre de Référence Déficits Immunitaires Héréditaires (CEREDIH).

We note that you have provided additional information within the Acknowledgements Section that is not currently declared in your Funding Statement. Please note that funding information should not appear in the Acknowledgments section or other areas of your manuscript. We will only publish funding information present in the Funding Statement section of the online submission form.Please remove any funding-related text from the manuscript and let us know how you would like to update your Funding Statement. Currently, your Funding Statement reads as follows: 

The author received no specific funding for this work

We have removed this part from the acknowledgments section.

This has been added within our cover letter

All study participants provided written informed consent. The study was performed in accordance with the 1975 Declaration of Helsinki and subsequent revisions, and was approved by the local institutional review board (CPP Ile de France II Paris, France), the French Advisory Committee on Data Processing in Medical Research (Paris, France). 

The full ethics statement has been added in the Materials and methods section (lines 183 to 186).

5. Review Comments to the Author

Reviewer #1: 

In the this study the authors present their observation that PD-L1 is expressed in activated human CD4+ T cells. They assess correlation expression levels of PD-L1 and other positive and negative costimulatory receptors such as ICOS, PD-1, CTLA4, TIGIT in T effector and Treg cells during culture of naïve CD4+ T cells with SEE superantigen and dendritic cells.

They found that activated T effector cells expressed BTLA, ICOS, PD-1 and PD-L1, whereas Treg cells expressed TIGIT, ICOS and CD27. They also observed that expression of PD-L1 in T effector cells positively correlated with their proliferation during exposure to SEE in the presence of DC. Titration experiments of Treg addition in the culture showed that suppression of T effector cell activation by the addition of Treg resulted in diminished expression of PD-L1. Based on these findings, the authors proposed that PD-L1 expression on T effector cells might serve as a marker of their activation and might be a useful biomarker in conditions of aberrant T cell activation such as in autoimmune diseases.

The results are interesting but not surprising based on our knowledge about the expression of these surface receptors in T cell subsets, including T effector cells and Treg.

Specific points:

1) Several previous studies have shown that PD-L1 is expressed in activated T cells (e.g. Bardhan et al. Sci Rep 2019; 9: 17252; Diskin et al. Nat Immunology 2020; 21: 442). Thus, this observation is not made for the first time as stated by the authors.

We agree with the reviewer as indeed an induced expression of PD-L1 has been described on CD3+CD4+ T cells isolated from PBMC of healthy donors after specific stimulation via TCR/CD3 and CD28 by using aCD3/aCD28/IgG coated beads (Bardhan et al. Sci Rep 2019; 9: 17252). Diskin et al. (Nat Immunology 2020; 21: 442) also described the induction of PD-L1 expression in T cells either in vivo in a mouse model (after Ag presentation in OVA-restricted CD4+ T cells) or in CD3/CD28 activated polyclonal CD4+ and CD8+ T cells from human PBMC and also on T cells derived from tumor. PD-L1 expression on T cells has been also described by another group (Daley et all. Cell 2016;166:1485). They showed that a unique activated �� T cells population expressing PD-L1 was the central regulators of effector T cell activation in pancreatic ductal adenocarcinoma (PDA) patients. These descriptions have been referred in the introduction (lines 106 to 110) and the discussion sections (lines 338 to 341) and the corresponding references were added in the references list. However, we would like to emphasize that the induction of PD-L1 expression on activated T cells (generated from SEE-stimulated naïve CD3+CD4+CD45RA+CD25negCD127+ human Teff cells, in the presence of autologous DCs) is an original description extending previous observations of PD-L1 expression on primary T cells. 

2) The authors correlated the expression of PD-L1 with the proliferation state of T effector cells. It is important to examine cytokine production by T cells under the same conditions.

Indeed, it would be very interesting to assess the production of cytokines by T cells under the same conditions. However, the numbers of APCs (2000) and T cells (8000 for effector T cells and 4000 for Treg) after sorting are too low to be able to perform this assay. The miniaturization of the Tregs suppressive assay was achieved to be applicable to the study of human blood samples from young patients with autoimmune disease and thus we cannot assess cytokines concentration in the very small volume of culture.

3) The studies as they stand are only correlative and the role of PD-L1 in regulating the function of T effector and Treg cells has not been examined. This aspect should be addressed by using appropriate antibodies targeting each of these surface receptors.

We fully agree with the reviewer that for the moment we are just making a correlation between PD-L1 expression and the regulation of T cell proliferation. Regarding the assessment of PD-L1 on T effector and Tregs function, it is not possible to achieve in our culture experiment setting. Indeed, at the initiation PD-L1 is expressed on DCs but not on naïve T cells. Adding an anti-PD-L1 antibody will primarily act on DCs and will thus perturb the T-DC interactions, but will not provide information on the role of PD-L1 on T cells function only.

4) All the experiments have been performed using SEE. It is well established that superantigens overcome the standard signals medicated by TCR and accessory molecules, including co-inhibitory and co-stimulatory receptors. For this reason, it would be more appropriate to assess the proposed role of these surface molecules of their interest in an antigen-specific system.

 We agree with the reviewer, but it is difficult in human model to use an antigen-specific system like the one used in mouse model (OVA model for instance). Of course, superantigen stimulation is a strong stimuli, but we were able to observe the Tregs suppressive functions. In contrast, when we used CD3/CD28-beads as a stimuli, the Tregs suppressive function were overcome. Therefore, even if the superantigen stimuli is stronger than an antigen-specific system, we found that it was weaker than the CD3/CD28 system and that it preserved the Tregs function in vitro. Of note, if the superantigens are used without DCs, we did not observe Teff proliferation, showing the requirement of the DCs in our system to activate naïve primary T cells.

---

## [Editor Report · Decision Letter 1]

5 Nov 2021

PD-L1 IS EXPRESSED ON HUMAN ACTIVATED NAIVE EFFECTOR CD4+ T CELLS. REGULATION BY DENDRITIC CELLS AND REGULATORY CD4+ T CELLS

PONE-D-21-22116R1

Dear Dr. Mazerolles,

We’re pleased to inform you that your manuscript has been judged scientifically suitable for publication and will be formally accepted for publication once it meets all outstanding technical requirements.

Kind regards,

Prof. Pierre Bobé

Academic Editor

PLOS ONE

---

## [Editor Report · Acceptance letter]

9 Nov 2021

PONE-D-21-22116R1 

PD-L1 is expressed on human activated naive effector CD4+ T cells.
Regulation by dendritic cells and regulatory CD4+ T cells 

Dear Dr. Mazerolles:

I'm pleased to inform you that your manuscript has been deemed suitable for publication in PLOS ONE. Congratulations! Your manuscript is now with our production department. 

Kind regards, 

on behalf of

Prof Pierre Bobé 

Academic Editor

PLOS ONE